# Impact of Hip Exercises on Postural Stability and Function in Patients with Chronic Lower Back Pain

**DOI:** 10.3390/diagnostics15101229

**Published:** 2025-05-13

**Authors:** Inhwan Leem, Gyu Bin Lee, Ji Won Wang, Sang Woo Pyun, Chang-Jun Kum, Jin Hyuck Lee, Hyeong-Dong Kim

**Affiliations:** 1Department of Physical Therapy, School of Health and Environmental Science, College of Health Science, Korea University, Seoul 02841, Republic of Korea; xy0709@korea.ac.kr; 2365 Maeil Korean Medicine Hospital, Seoul 02582, Republic of Korea; anhubris@naver.com; 3Department of Sports Medical Center, Anam Hospital, College of Medicine, Korea University, Seoul 02841, Republic of Korea; humanwellness@naver.com (G.B.L.); jiwon-eee@naver.com (J.W.W.); pyun3333@gmail.com (S.W.P.)

**Keywords:** chronic non-specific lower back pain, restricted hip extension mobility, modified Thomas test, postural stability, Oswestry disability index

## Abstract

**Background and Objective:** This study aimed to compare functional and clinical outcomes between patients with chronic non-specific lower back pain (NSLBP) with restricted hip extension mobility who performed spinal stabilization exercises with hip mobilization either with or without additional hip exercises. **Methods:** A total of 42 patients with chronic NSLBP with restricted hip extension mobility were enrolled (21 with and 21 without additional hip exercises). Functional and clinical outcomes were assessed based on hip joint mobility, back extensor endurance, postural stability, and patient-reported outcomes, including visual analog scale (VAS) and Oswestry disability index (ODI) scores. **Results:** A significant group–time interaction was identified for postural stability (Rt: *p* < 0.001, Lt: *p* = 0.002) and the ODI (*p* = 0.004). After the intervention, the group with additional hip exercises demonstrated significantly greater improvements in postural stability (Rt: *p* < 0.001; Lt: *p* = 0.01) and ODI scores (*p* < 0.001) compared with the group without additional hip exercises. However, no significant main group effect or group–time interaction was observed for the hip joint mobility, back extensor endurance, or VAS scores (all *p* > 0.05). Furthermore, the ODI score (r^2^ = 0.123, *p* = 0.023) was an independent predictor of hip joint mobility but not the VAS score (*p* > 0.05). **Conclusions:** Hip exercises may improve postural stability and function in patients with chronic NSLBP with restricted hip extension mobility. Notably, clinicians and therapists must recognize the importance of hip exercises during rehabilitation of patients with chronic NSLBP with restricted hip extension mobility.

## 1. Introduction

Lower back pain (LBP) occurs in 80% of the general population at least once during a person’s lifetime [1], with the highest prevalence rate being in the 40–80 age group. Despite advances in imaging and diagnostic methods [2], 90% of non-specific LBP (NSLBP) cases are classified as chronic LBP [3], which is clinically challenging to diagnose because of its unidentifiable pathology. Potential contributing factors to NSLBP include weakness of the core muscles [4,5], weakness of the hip muscles [6,7], restricted hip joint mobility [8,9], poor postural stability [10,11], and muscle fatigue [12], among the various causes. Therefore, many physiotherapists focus on strengthening the core and hip muscles, improving postural stability, and restoring hip joint mobility to treat NSLBP. However, their benefits remain unclear.

Several systematic reviews have reported that exercise interventions effectively reduce pain and improve physical function in patients with chronic NSLBP [13,14]. Specifically, spinal stabilization exercises can relieve pain and improve back muscle strength while stabilizing the spine in patients with chronic NSLBP [4]. Furthermore, Aoyagi et al. [15] reported that spinal mobilization was effective in improving the pain, range of motion (ROM), and function in patients with chronic NSLBP. However, clinicians often encounter patients with chronic NSLBP with restricted hip extension mobility due to tight hip flexors. This restriction leads to increased compensatory lumbar spine motion [9,16,17], increasing pressure on the lumbar spine and thereby contributing to LBP. Notably, restricted hip extension mobility due to tight hip flexors can weaken the hip muscles [17,18], further increasing the stress on the lumbar spine and causing LBP [7]. Sembrano and Polly reported that approximately 12.5% of patients with chronic LBP have an underlying hip joint pathology [19]. Therefore, considering the biomechanical relationship between the hip joint and lumbar spine, hip joint dysfunction may contribute to chronic NSLBP [19,20,21,22]. However, hip joint diseases [23,24], such as avascular necrosis of the femoral head caused by an impaired blood supply, are often misdiagnosed as LBP, as the pain is localized in the lower back along with limited hip mobility [23]. Therefore, to the best of our knowledge, no studies have directly compared the functional and clinical outcomes between patients with chronic NSLBP with restricted hip extension mobility with or without performing additional hip exercises. Consequently, the functional and clinical effectiveness of hip exercises in these patients remains unclear.

This study aimed to compare the functional and clinical outcomes, including hip joint mobility, back extensor endurance, postural stability, and patient-reported out-comes, between patients with chronic NSLBP with restricted hip extension mobility who performed spinal stabilization exercises with hip mobilization, both with and without additional hip exercises. We hypothesized that additional hip exercises would yield better outcomes in patients with chronic NSLBP with restricted hip extension mobility.

## 2. Methods

This prospective comparative study adhered to the CONSORT guidelines for non-pharmacological treatments and was conducted in accordance with the Declaration of Helsinki. The study was approved by the Institutional Review Board of Korea University (IRB No: 2024-0241-01). Written informed consent was obtained from all participants, and their rights were duly protected.

### 2.1. Patient Enrollment

Seventy patients with chronic LBP (pain for >3 months) participated in this study between July 2024 and January 2025. To determine eligibility, all diagnoses were confirmed via radiography and physical examination of the spine and hip joints. Patients aged 40–60 years were included in this study if they had chronic NSLBP with restricted hip extension mobility, as assessed using the modified Thomas test (MTT). Specifically, 28 patients were excluded from the study for the following reasons: normal hip mobility; neurological symptoms; lumbar fracture; hip joint diseases; scoliosis; prior ankle, knee, or spine surgery; or injections within the past 3 months. Finally, 42 patients with chronic NSLBP with restricted hip extension were enrolled (Figure 1), including 21 in the spinal stabilization exercises + hip mobilization + hip exercises group (group with additional hip exercises) and 21 in the spinal stabilization exercises + hip mobilization group (group without additional hip exercises).

### 2.2. Outcome Measures

#### 2.2.1. Hip Joint Mobility

The MTT was used to assess restricted hip extension (Figure 2A) [9,16]. A positive value (restricted hip extension) was recorded when the knee was positioned above the hip, and a negative value (normal hip joint mobility) was recorded when the knee was below the hip. In this study, the intraclass correlation coefficient (ICC) for the MMT was 0.87.

#### 2.2.2. Back Extensor Endurance

The Ito test was used to assess the isometric back extensor endurance [25]. In the prone position, the sternum was lifted off the floor, with a small pillow placed under the lower abdomen (Figure 2B). The holding duration(s) was recorded at a maximum limit of 300 s to prevent back muscle fatigue [25]. In this study, the ICC for the Ito test was 0.81.

#### 2.2.3. Postural Stability

The Trendelenburg test, a one-leg standing test, was used to evaluate static balance [26]. Patients were instructed to close their eyes, place both hands on their hips, and maintain balance on one leg (Figure 2C). The time required for the foot to touch the ground was recorded. In this study, the ICC for the Trendelenburg test was 0.84.

#### 2.2.4. Pain and Function (Disability)

Pain and function (disability) were measured using the visual analog scale (VAS) and Oswestry disability index (ODI), respectively [27]. The VAS score ranges from 0 to 10 points, with 0, 4, and 10 indicating no pain, moderate pain, and severe pain, respectively. The ODI comprises 10 items, including pain intensity, personal care, lifting, walking, sitting, standing, sleeping, sex life, social life, and traveling, with 0% and 100% indicating no disability and the highest level of disability, respectively. In this study, the ICC for the VAS and ODI was 0.93 and 0.88, respectively.

#### 2.2.5. Interventions

In this study, 42 patients with chronic NSLBP with restricted hip extension were randomly allocated to two groups (1:1 ratio). The intervention involved hip mobilization, spinal stabilization exercises, and hip exercises.

Hip mobilization was performed to improve the hip joint ROM. To improve hip extension and external rotation in the prone position, the femoral head was glided from the posterior to the anterior by applying pressure from the palm for 10 min (posterior-to-anterior glide, Grade II). Patients performed spinal stabilization exercises, including side-bridge, bird-dog, and Pallof press exercises, to strengthen muscles such as the erector spinae, abdominis, and quadratus lumborum and improve lower back stability. In the additional hip exercises group, hip exercises, including the supine bridge, clamshell, and hip abduction in a single-leg standing position, were performed to improve the strength of gluteal muscles, including the gluteus medius and maximus. During all exercises, patients maintained a neutral position through the abdominal bracing techniques. Each exercise was performed in two sets 10–15 times with a 1–2 min rest interval between sets. The difficulty of the exercises was adjusted according to pain intensity. The intervention was conducted 3 days a week for 6 weeks at the institution, whereas the home exercise program was instructed to be performed daily, except on institution visit days. Adherence to the home exercise program was monitored with an exercise diary. These exercises focused on improving the stability and strength of the lumbar and hip joints and were expected to help reduce pain and improve function in patients with chronic LBP. Detailed descriptions of these exercises are provided in Appendix A.

### 2.3. Statistical Analysis

Based on a previous study utilizing the VAS in patients with chronic NSLBP [28], a VAS score difference of >2 points between the spinal stabilization exercises and conventional exercises (including stretching and muscle strengthening exercises) was considered clinically significant. A priori power analysis was performed to determine the sample size, which was calculated to be 22 (11 per group) for an alpha level of 0.05 and a power of 0.8. The power to detect between-group differences in the VAS scores was 0.812.

The Shapiro–Wilk test was performed to determine the normal distribution of continuous variables. Independent *t*-tests or chi-square tests were used to compare demographic information between the two groups. We analyzed the data using a mixed effects model, taking into account within-subject correlations and individual variability. Participants were included as a random effect to provide more accurate estimates of the fixed effects. When significant interactions were observed, Tukey-adjusted *p*-values were used for pairwise comparisons between groups and across pre- and post-intervention time points. Pearson’s correlation analysis was used to confirm the correlations among hip joint mobility, back extensor endurance, postural stability, VAS scores, and ODI scores. For factors associated with the Pearson correlation analysis, multiple linear regression analysis was performed to identify the influence of the predictor variables on the dependent variable (hip joint mobility). Statistical analyses were performed using IBM^®^ SPSS^®^ Statistics 20 (SPSS Inc., Chicago, IL, USA), and statistical significance was set at *p* < 0.05.

## 3. Results

### 3.1. Demographic Data

Table 1 presents the demographic data of the participants. No significant differences were observed in sex, age, height, weight, or body mass index between the two groups (*p* > 0.05).

### 3.2. Between- and Within-Group Comparisons of Functional and Clinical Outcomes

A group main effect was observed for postural stability (Rt: *p* = 0.006, Lt: *p* = 0.046) and the ODI (*p* = 0.023), indicating better postural stability and lower disability in the group with additional hip exercises. A time main effect was observed for hip joint mobility (*p* < 0.001), back extensor endurance (*p* < 0.001), postural stability (*p* < 0.001), VAS score (*p* < 0.001), and the ODI (*p* < 0.001), indicating all parameters significantly improved from pre- to post-intervention in both groups (Table 2 and Table 3). Furthermore, a significant group–time interaction was identified for postural stability (Rt: *p* < 0.001, Lt: *p* = 0.002) and the ODI (*p* = 0.004). After the intervention, significant between-group differences were observed in postural stability (Rt: 20.0 ± 1.6 s vs. 16.1 ± 1.6 s, 95% confidence interval [CI]: 1.9–4.9, *p* < 0.001; Lt: 18.2 ± 2.1 s versus 15.8 ± 3.6 s, 95% CI: 0.6–4.3, *p* = 0.01; Table 2) and the ODI (12.3 ± 2.8 versus 18.4 ± 3.5, 95% CI: from −8.1 to −4.1, *p* < 0.001; Table 3), indicating greater improvements in postural stability and functional ability in the group with additional hip exercises compared with the group without additional hip exercise. However, no significant group main effect or group–time interaction was observed for the hip joint mobility, back extensor endurance, and VAS scores (all *p* > 0.05).

### 3.3. Correlation and Predictor Factors

Correlation analyses were performed between hip joint mobility, back extensor endurance, postural stability, VAS score, and ODI score. Bivariate analysis revealed that the VAS (r = 0.351, *p* = 0.023) and ODI (r = 0.499, *p* = 0.001) scores were significantly correlated with hip joint mobility. Multiple linear regression analysis of these three parameters revealed that the ODI score (r^2^ = 0.123, *p* = 0.023) was an independent predictor of hip joint mobility but not the VAS score (r^2^ = 0.022, *p* = 0.734; Table 4).

## 4. Discussion

This study aimed to compare functional and clinical outcomes, including hip joint mobility, back extensor endurance, postural stability, and patient-reported outcomes, between patients with chronic NSLBP with restricted hip extension mobility who performed spinal stabilization exercises with hip mobilization with and without additional hip exercises. The most important finding of this study was that in patients with chronic NSLBP with restricted hip extension mobility, additional hip exercises along with spinal stabilization exercises with hip mobilization better improved postural stability and functional ability compared with patients performing spinal stabilization exercises with hip mobilization alone. However, hip joint mobility, back extensor endurance, and VAS scores did not exhibit significant differences. Furthermore, the ODI score was closely associated with hip joint mobility.

A previous study reported that the hip flexor muscles affect lumbar spine stability [29]. However, tight hip flexors cause chronic LBP [17,30], which also contributes to isometric back muscle weakness [31]. Therefore, joint mobilization [15] and exercise therapy [32] can be effective in recovering the restricted ROM and improving muscle strength in patients with chronic NSLBP. Specifically, hip joint mobilization may elicit a neurophysiological response along with muscle stretching to relieve pain [22]. In addition, spinal stabilization exercises may reduce pain by improving lumbar stability and strengthening back muscles during upper and lower extremity movements [33]. Consequently, as spinal stabilization exercises and hip mobilization were applied in both groups in this study, no differences were observed in hip joint mobility, back extensor endurance, or pain between the groups. Hence, in patients with chronic NSLBP, adding hip exercises may not lead to significant differences in hip joint mobility, back extensor endurance, and pain.

A meta-analysis [34] and systematic review [35] reported poor postural stability in patients with chronic LBP, which may be associated with weakened hip muscles [36]. In particular, reduced hip abductor strength, specifically in the gluteus medius, is noticeable in patients with chronic NSLBP [37]. Hip abductors stabilize the pelvis in the frontal plane during a single-leg stance [38,39]. Therefore, hip exercises may play a crucial role in improving postural stability [40]. Brech et al. [41] evaluated postural stability, pain, and disability in patients with chronic NSLBP and reported an inverse correlation between postural stability and ODI scores. In addition, Ostelo et al. [42] reported that a 30% improvement in the ODI represents a useful threshold for identifying clinically meaningful improvements in patients with chronic LBP. In the present study, despite the improvement in both functional and clinical outcomes after the intervention in both groups, the patients with chronic NSLBP who performed additional hip exercises exhibited better postural stability (*p* < 0.001) and ODI scores (*p* < 0.001) than those who did not. However, only the group with additional hip exercises exhibited 30% improvement in ODI scores, which was also closely associated with hip joint mobility. Accordingly, our results indicate that hip exercises may be beneficial for improving postural stability, functional ability, and hip joint mobility in patients with chronic NSLBP. Kim and Shin [43] reported a strong correlation between limited hip extension and ODI scores in patients with chronic NSLBP. Our results corroborate these previous findings. However, postural stability may be affected by various factors, such as proprioception [44], psychological [45], and visual factors [46]. Therefore, high-quality studies, including prospective randomized controlled studies, are essential to validate our findings regarding the effectiveness of hip exercises in patients with chronic NSLBP.

### Limitations

This study also has some limitations. First, owing to the small sample size and lack of a normal control group, the results of this study cannot be generalized. Therefore, studies with larger sample sizes are warranted to confirm these findings. Second, this study only examined the short-term follow-up and lacked assessor blinding. Consequently, research on medium- and long-term follow-up with assessor blinding is necessary. Third, we did not evaluate the reduction in hip joint space. Ulusoy and Kıvrak [47] reported that a reduction in hip joint space may increase the risk of hip osteoarthritis. Thus, loss of hip mobility due to further degenerative changes may contribute to LBP. Fourth, we did not assess potential confounding variables, such as baseline health status or prior training, which may have influenced the outcomes. Therefore, further studies controlling for these variables are required. Finally, as the spinal stabilization exercises in this study also included hip exercises, a more precisely designed exercise program is necessary to clarify the study results. Despite these limitations, to the best of our knowledge, this is the first study to assess the effect of incorporating additional hip exercises into spinal stabilization exercises with hip mobilization on functional and clinical outcomes in patients with chronic NSLBP with restricted hip extension mobility.

## 5. Conclusions

This study revealed that hip exercises may improve postural stability and function in patients with chronic NSLBP with restricted hip extension mobility. Therefore, clinicians and therapists should consider hip exercises during the rehabilitation of patients with chronic NSLBP with restricted hip extension mobility.

## Figures and Tables

**Figure 1 diagnostics-15-01229-f001:**
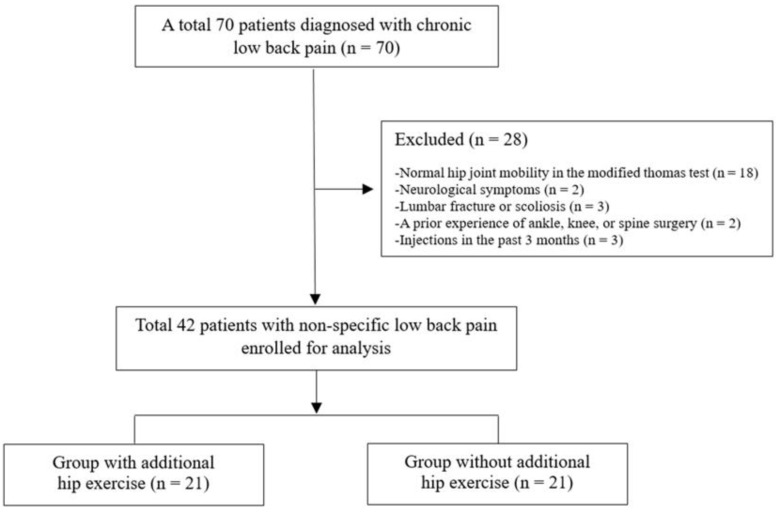
Study flowchart.

**Figure 2 diagnostics-15-01229-f002:**
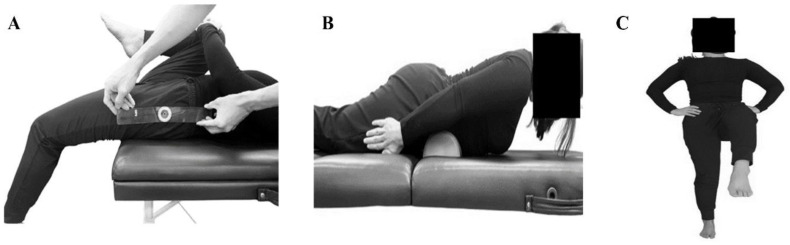
(**A**) Hip joint mobility, (**B**) back extensor endurance, and (**C**) postural stability tests.

**Table 1 diagnostics-15-01229-t001:** Demographic characteristics of the participants.

	Group with Additional Hip Exercises (n = 21)	Group without Additional Hip Exercises (n = 21)	*p*-Value
Sex (male/female)	8/13	9/12	1.0
Age (years) ^a^	47.5 ± 4.5	45.6 ± 3.9	0.138
Height (cm) ^a^	166.9 ± 9.2	168.5 ± 7.5	0.560
Weight (kg) ^a^	61.3 ± 12.8	63.6 ± 9.7	0.527
Body mass index (kg/m^2^) ^a^	19.6 ± 2.4	20.1 ± 2.0	0.521

^a^ Values are expressed as the mean ± standard deviation.

**Table 2 diagnostics-15-01229-t002:** Comparison of functional outcomes.

		Group with Additional Hip Exercises (n = 21)	Group Without Additional Hip Exercises (n = 21)	*p*-Value ^1^
Hip joint mobility (right)	Pre-intervention	18.0 ± 6.4	16.0 ± 6.9	0.187
	*MD (95% CI), ES*	2.0 (−1.4–6.9), 0.148		
	Post-intervention	−5.9 ± 2.3	−5.9 ± 1.5	0.943
	*MD (95% CI), ES*	0 (−1.2–1.2), 0		
	*p*-value ^2^	**<0.001**	**<0.001**	
Hip joint mobility (left)	Pre-intervention	15.0 ± 7.9	14.0 ± 6.8	0.605
	*MD (95% CI), ES*	−1.0 (−3.4–5.8), 0.067		
	Post-intervention	−7.9 ± 3.4	−6.6 ± 3.2	0.209
	*MD (95% CI), ES*	−1.3 (−3.3–0.7), −0.193		
	*p*-value ^2^	**<0.001**	**<0.001**	
Back extensor endurance	Pre-intervention	88.2 ± 10.2	85.2 ± 8.7	0.313
	*MD (95% CI), ES*	3.0 (−2.9–8.9), 0.156		
	Post-intervention	123.6 ± 15.3	120.3 ± 9.4	0.413
	*MD (95% CI), ES*	3.3 (−4.7–11.1), 0.128		
	*p*-value ^2^	**<0.001**	**<0.001**	
Postural stability (right)	Pre-intervention	8.7 ± 1.5	9.5 ± 1.7	0.147
	*MD (95% CI), ES*	−0.8 (−1.8–0.3), −0.242		
	Post-intervention	20.0 ± 1.6	16.1 ± 1.6	**<0.001**
	*MD (95% CI), ES*	3.9 (1.9–4.9), 0.773		
	*p*-value ^2^	**<0.001**	**<0.001**	
Postural stability (left)	Pre-intervention	9.0 ± 1.3	9.4 ± 1.5	0.305
	*MD (95% CI), ES*	−0.4 (−1.3–0.4), −0.141		
	Post-intervention	18.2 ± 2.1	15.8 ± 3.6	**0.01**
	*MD (95% CI), ES*	2.4 (0.6–4.3), 0.377		
	*p*-value ^2^	**<0.001**	**<0.001**	

MD = mean difference; CI = confidence interval; ES = effect size. Note: The measurement units of hip joint mobility, back extensor endurance, and postural stability are expressed in degrees (°), seconds, and seconds, respectively. Here, *p*-value ^1^ = comparison between two groups; *p*-value ^2^ = comparison within each group. Values are expressed as the mean ± standard deviation.

**Table 3 diagnostics-15-01229-t003:** Comparison of clinical outcomes.

		Group with Additional Hip Exercises (n = 21)	Group Without Additional Hip Exercises (n = 21)	*p*-Value ^1^
VAS	Pre-intervention	4.3 ± 0.7	4.1 ± 0.7	0.379
	*MD (95% CI), ES*	0.2 (−0.2–0.6), 0.141		
	Post-intervention	2.1 ± 0.7	2.2 ± 0.7	0.657
	*MD (95% CI), ES*	−0.1 (−0.5–0.3), −0.071		
	*p*-value ^2^	**<0.001**	**<0.001**	
ODI	Pre-intervention	43.0 ± 8.8	41.1 ± 8.4	0.488
	*MD (95% CI), ES*	1.9 (−3.5–7.2), 0.109		
	Post-intervention	12.3 ± 2.8	18.4 ± 3.5	**<0.001**
	*MD (95% CI), ES*	−6.1 (from −8.0 to −4.1), −0.693		
	*p*-value ^2^	**<0.001**	**<0.001**	

VAS = visual analog scale; ODI = Oswestry disability index; MD = mean difference; CI = confidence interval; ES = effect size. VAS and ODI measurement units are expressed as points and percentages (%), respectively. Here, *p*-value ^1^ = comparison between two groups; *p*-value ^2^ = comparison within each group. Values are expressed as the mean ± standard deviation.

**Table 4 diagnostics-15-01229-t004:** Multiple linear regression analysis results of the predictors of hip joint mobility.

Dependent Variables	Independent Variables		Standardized Coefficients		
		*t*	β	*r* ^2^	*p*-value
Hip joint mobility	VAS	0.342	0.054	0.022	0.734
	ODI	2.367	0.351	0.123	**0.023**

VAS = visual analog scale; ODI = Oswestry disability index.

## Data Availability

Data presented in this study are available upon request from the corresponding author.

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
