# Peer review of "Impact of Hip Exercises on Postural Stability and Function in Patients with Chronic Lower Back Pain"

_diagnostics, 2025, doi:10.3390/diagnostics15101229_

Round 1

Reviewer 1 Report

Comments and Suggestions for Authors

  1. The title is too long and complex; consider shortening it for better readability, for example: “Impact of Hip Exercises on Postural Stability and Function in Chronic Low Back Pain Patients.”
  2. The abstract includes excessive methodological detail; it should be more concise, and the conclusion should be stronger and clearer, e.g., “Hip exercises improve function and stability in NSLBP patients.”
  3. The introduction is overly lengthy at times; sentences should be shortened for clarity, and the hypothesis should be clearly stated, e.g., “We hypothesized that additional hip exercises would yield better outcomes.”
  4. A brief discussion of hip joint diseases’ etiology would strengthen the biomechanical rationale of the study; consider mentioning that avascular necrosis of the femoral head results from disrupted blood supply (doi:10.1186/s13018-023-04297-0).
  5. The methodology section should clearly describe the hip mobilization technique (e.g., posterior-to-anterior glide, Grade II) and how adherence to the home exercise program was monitored (e.g., exercise diaries).
  6. p-values should be written according to English scientific writing conventions (e.g., p = 0.010), and abbreviations such as “MD” should be explicitly defined in all tables (e.g., MD: Mean Difference).
  7. The authors should briefly mention that reduction in hip joint space may increase the risk of hip osteoarthritis, thus linking hip mobility loss to further degenerative changes (doi:10.1186/s13018-023-03932-0).
  8. In the discussion, findings should be presented more directly, and the clinical significance of changes (e.g., ODI improvement thresholds) should be emphasized.
  9. In the limitations section, it is important to mention the lack of assessor blinding and the short follow-up period.
  10. The conclusion should be rewritten with more scientific and neutral phrasing, e.g., “Our findings suggest that hip exercises may enhance rehabilitation outcomes.”
  11. Minor grammatical errors throughout the text should be corrected (e.g., “colosely” → “closely”), and reference formatting should be standardized (e.g., “[13, 14]” instead of “[13,14]”).

Author Response

Thank you for the review and comments. We have done our best to respond to your comment.

Reviewer 2 Report

Comments and Suggestions for Authors

This manuscript studies the effects of hip exercises on postural stability and function in patients with chronic non-specific low back pain (LBP) who have restricted hip extension mobility. While the study’s aim is precise and addresses an important question in the musculoskeletal rehabilitation field, significant issues with scientific methodology and statistical analysis must be addressed. These issues lower the robustness and clarity of the study’s conclusions.

Major Issues

- Is the sample size (n=12) sufficient for generalizing the results? Statistical power analysis must be conducted to ensure the detection of meaningful effects. Small sample sizes pose a risk of Type II errors and affect the external validity of the findings. The authors should justify the sample size with a power analysis and consider increasing it if possible.

- How did the authors control for potential confounding variables that may influence the outcomes? Specifically, participant characteristics (such as age, baseline health status, or prior training) may influence the results, but these factors are not adequately accounted for. The authors should either statistically control for these confounders (e.g., by including them as covariates in their analysis) or provide more detailed information.

- Please provide validity and reliability of all measurements taken; otherwise, this limits the reproducibility of the results.

- Lines 123 - 135: The description of the intervention is vague. How were participants assigned to the intervention groups? Also, how was the intervention administered and monitored? This lack of clarity raises concerns about the consistency and standardization of the intervention across participants. The authors should provide a more detailed description of the intervention, including the protocol followed and any adherence checks performed during the study.

- The statistical analyses are not entirely appropriate for the data. Given the within-subjects design, a mixed-effects model would be more suitable than relying solely on other methods, such as ANOVA. This would better account for the repeated measures nature of the data and individual participant variability. Therefore, the authors should consider using mixed-effects models or appropriate statistical techniques.

- How did the authors handle multiple comparisons? This could lead to an increased risk of Type I errors. If multiple tests are being conducted, adjustments (e.g., Bonferroni or false discovery rate correction) should be applied. Please clearly state how the multiple comparisons were controlled for.

- Finally, please report statistical power calculations and effect sizes for the main analyses to clarify the magnitude and reliability of the observed effects. This information is essential to understanding the practical significance of the findings.

Despite the statistical and methodological concerns, the preliminary findings provide useful insights into the topic and suggest that the intervention may have some impact, warranting further research.

Author Response

(The authors gave the same response as above.)

Reviewer 3 Report

Comments and Suggestions for Authors

I read the paper by Inhwan Leem et al. and congratulate them for demonstrating the benefits of incorporating some specific hip exercises in rehabilitation programs for patients with chronic non-specific low back pain and limited hip extension mobility. The study shows that adding hip exercises to spinal stabilization improves postural stability and functional ability. These exercises help reduce pain and improve mobility, and also contribute to sustainable recovery, emphasizing personalized rehabilitation strategies for better patient outcomes.

 ​The abstract summarizes the primary ideas in the work, and the keywords are appropriately selected. The introduction provides sufficient background to understand the initial context. Hip joint mobility is connected to low back pain (LBP) due to the biomechanical link between the hip and lumbar spine. Limited hip extension mobility, often caused by tight hip flexors, can result in compensatory movements in the lumbar spine, increasing stress and pressure on the lower back. This contributes to chronic non-specific low back pain (NSLBP).

The study design is clear, and the methods are well explained. I have a few recommendations.

The inclusion and exclusion criteria necessitate to be more precise. Although the highest prevalence rate is documented in the literature for individuals aged 40 to 80, this study primarily targets patients at the lower end of this spectrum. This focus may indicate selection bias and restrict the generalizability of the findings to the broader population with NSLBP. It would be advisable to specify a narrower age range within the inclusion criteria.

Hip disease should be ruled out with a medical examination. Additionally, specific spine diseases beyond lumbar fracture or scoliosis should be excluded, as they may affect exercise difficulty.

Furthermore, the allocation method of the 42 patients into the two study groups was unclear.

Regarding the interventions, the authors are encouraged to provide brief descriptions for intervention exercises, including references to involved muscles and anticipated outcomes. 

Spinal stabilization exercises in the study also included hip exercises, which may have confounded the results. ​ This overlap makes it difficult to isolate the specific effects of the additional hip exercises.  A more precisely designed exercise program is needed to isolate the effects of additional hip exercises. ​The authors already mentioned these limitations; I recommend a more accurate description of the study objectives and interventions would benefit readers.

The results are comprehensively presented and support the conclusions effectively. The discussions emphasize the biomechanical relationship between the spine and hip, highlighting the importance of considering this segment in a unified manner.

Author Response

(The authors gave the same response as above.)

Round 2

Reviewer 2 Report

Comments and Suggestions for Authors

I appreciate the authors’ effort to acknowledge all suggestions. The manuscript’s quality improved greatly, but one challenge remains (question 5) about the statistical analysis method used. 

1) While I understand that the primary focus of the study is on post-intervention group differences, the justification for using a two-way repeated-measures ANOVA rather than a mixed-effects model remains unconvincing. The study involves repeated measures data, and even if baseline differences were non-significant, the variability across time points and individual participants still warrants consideration. A mixed-effects model would not only account for within-subject correlations and individual variability but also provide a more comprehensive analysis of the data structure without relying on multiple post hoc tests. It also handles missing data more effectively than ANOVA.

2) Additionally, the use of post hoc t-tests to explore significant effects is inappropriate in the context of a repeated-measures design. The more appropriate approach would be to conduct simple main effects analyses to decompose significant interactions, which maintains the integrity of the repeated measures structure without inflating the risk of type I errors.

Therefore, I strongly recommend that the authors reconsider their analytical approach, either by implementing a mixed-effects model or by providing a more robust justification for the chosen ANOVA framework, clearly addressing the limitations of post hoc t-tests and the appropriate use of simple main effects analysis. 

Nevertheless, I find that this challenge occurred due to unclear aim, and, hence, analytical focus. The stated aim emphasizes functional and clinical outcomes, including hip joint mobility, back extensor endurance, postural stability, and patient-reported outcomes across intervention groups. However, it is unclear whether the focus is on comparing post-intervention outcomes across groups, assessing within-subject changes over time, or both. This lack of clarity directly impacts the choice of statistical analysis.

Therefore, clearly specify whether the primary objective is to assess between-group differences at post-intervention, within-subject changes over time, or the interaction of both. However, even though the authors suggested that they focused primarily on post-intervention group differences, implying that within-subject effects were not considered relevant, it is a fundamental misinterpretation of the study design. In a repeated-measures design, within-subject effects are inherently relevant because they capture individual variability and temporal changes, which are crucial for understanding intervention effects. Moreover, ignoring within-subject effects not only underutilizes the data but also increases the risk of overlooking meaningful individual or time-dependent effects. This is particularly important given that the outcomes include functional and clinical measures that likely vary both across time and between groups.

Author Response

Thank you for your comment.
Please see attached file
